# Effect of DNA Aptamer Concentration on the Conductivity of a Water-Gated Al:ZnO Thin-Film Transistor-Based Biosensor

**DOI:** 10.3390/s22093408

**Published:** 2022-04-29

**Authors:** Andrejs Ogurcovs, Kevon Kadiwala, Eriks Sledevskis, Marina Krasovska, Ilona Plaksenkova, Edgars Butanovs

**Affiliations:** 1Institute of Solid State Physics, University of Latvia, Kengaraga Street 8, LV-1063 Riga, Latvia; kevon.kadiwala@cfi.lu.lv (K.K.); edgars.butanovs@cfi.lu.lv (E.B.); 2G. Liberts’ Innovative Microscopy Centre, Department of Technology, Institute of Life Sciences and Technology, Daugavpils University, Parades Street 1A, LV-5401 Daugavpils, Latvia; eriks.sledevskis@du.lv (E.S.); marina.krasovska@du.lv (M.K.); 3Laboratory of Genomics and Biotechnology, Department of Biotechnology, Institute of Life Sciences and Technology, Daugavpils University, Parades Street 1A, LV-5401 Daugavpils, Latvia; ilona.plaksenkova@du.lv

**Keywords:** biosensor, zinc oxide, thin-film transistor, DNA, electrochemistry

## Abstract

Field-effect transistor-based biosensors (bio-FETs) are promising candidates for the rapid high-sensitivity and high-selectivity sensing of various analytes in healthcare, clinical diagnostics, and the food industry. However, bio-FETs still have several unresolved problems that hinder their technological transfer, such as electrical stability. Therefore, it is important to develop reliable, efficient devices and establish facile electrochemical characterization methods. In this work, we have fabricated a flexible biosensor based on an Al:ZnO thin-film transistor (TFT) gated through an aqueous electrolyte on a polyimide substrate. In addition, we demonstrated techniques for establishing the operating range of such devices. The Al:ZnO-based devices with a channel length/width ratio of 12.35 and a channel thickness of 50 nm were produced at room temperature via magnetron sputtering. These Al:ZnO-based devices exhibited high field-effect mobility (μ = 6.85 cm^2^/Vs) and threshold voltage (V_th_ = 654 mV), thus showing promise for application on temperature-sensitive substrates. X-ray photoelectron spectroscopy was used to verify the chemical composition of the deposited films, while the morphological aspects of the films were assessed using scanning electron and atomic force microscopies. The gate–channel electric capacitance of 40 nF/cm^2^ was determined using electrochemical impedance spectroscopy, while the electrochemical window of the gate–channel system was determined as 1.8 V (from −0.6 V to +1.2 V) using cyclic voltammetry. A deionized water solution of 10 mer (CCC AAG GTC C) DNA aptamer (molar weight −2972.9 g/mol) in a concentration ranging from 1–1000 pM/μL was used as an analyte. An increase in aptamer concentration caused a proportional decrease in the TFT channel conductivity. The techniques demonstrated in this work can be applied to optimize the operating parameters of various semiconductor materials in order to create a universal detection platform for biosensing applications, such as multi-element FET sensor arrays based on various composition nanostructured films, which use advanced neural network signal processing.

## 1. Introduction

Biosensors are analytical devices that convert changes in the physical or chemical properties of a biological receptor (biomatrix) into electrical or other types of signals, the amplitude of which depends on the concentration of the analytes [1,2]. Biosensors have been designed to replace the expensive, complex, and time-consuming procedures of classical bioanalytics. A necessary property of the biomatrix is its selectivity for a given analyte, such as enzymes [3,4], antibodies [5,6], and single-stranded DNA sequences (aptamers) [7]. Biosensors (immunosensors) have a diverse range of applications in areas such as healthcare [8,9], clinical diagnostics [10,11], and food sensing [12,13,14]. The most commonly used are optical [15,16], electrochemical [17,18], and field-effect transistor (FET) [19,20] aptasensors. The widespread use of these types of sensors is due to the fact that the monoclonal antibodies obtained by the method proposed by Köchler and Milstein [21] have an almost inexhaustible variety and can be produced under the influence of a huge range of antigens. This, in turn, enables functionalization of the sensor, depending on the scope of use and the composition of the analyte.

In the process of developing a sensitive biosensor, an important step is the immobilization of the biomatrix molecules on the transistor working surface, which can be effectively increased by using nanostructured films. Materials of various composition, such as metal oxides, have been used as platforms for immobilization. In particular, zinc-oxide-based immunosensors have been developed for the detection of important biological markers such as α-1-fetoprotein (AFP) [22], and C-reactive protein (CRP) [23]. Zinc oxide (ZnO) has physical and chemical properties that are necessary for bioanalytical applications: it is nontoxic, chemically stable, and electrochemically active. The biocompatibility of ZnO has been shown in several studies using cytotoxicity tests [24] and hemolysis [25,26]. The negative effect of ZnO nanoparticles on living cells begins to appear only at sufficiently high concentrations (about 100 µg/mL) [27], which enables the use of ZnO structures for bioanalytical measurements in vivo. The high isoelectric point of ZnO (9.5) allows the attachment of substances with low values of the isoelectric point, which enables the immobilization of enzymes on ZnO by simple physical adsorption. The transport properties of zinc oxide (specific resistivity, mobility, charge carrier concentration, and transistor on-to-off current ratio) can vary over a wide range [28], depending on the synthesis method or doping, for example, with aluminum (Al:ZnO) [29,30].

Because of their high sensitivity to external electric fields, thin-film transistors (TFTs) can be effectively used as sensitive elements in various diagnostic equipment [31]. While a gate electrode is commonly used to modulate the electronic response of the channel, placing chemical compounds of biological origin on the surface of a TFT enables the development of a new class of devices called field-effect transistor-based biosensors, also known as bio-FETs [32]. The principle of the operation of a bio-FET is shown in Figure 1a. To increase selectivity and sensitivity, the TFT channel undergoes a functionalization process, i.e., the placement and attachment of receptors, such as amino acids, antibodies, and enzymes, on the surface of the channel by adsorption or linker molecules [33]. When the target molecule binds to a bioreceptor (e.g., glucose, hemoglobin, cholesterol, and hydrogen peroxide) on the sensor surface through covalent bonding, electrostatic, or Van der Waals forces, it results in a change in net electric charge in the semiconductor channel. Properly functionalized bio-FET can detect target analytes with high sensitivity and selectivity [34]. The use of Al:ZnO as the semiconductor layer in an electrolytically controlled transistor is of particular interest due to the low operating voltages involved (<1 V). In such devices, the semiconductor comes into direct contact with the aqueous medium without requiring prior encapsulation of the transistor elements, leading to the formation of an electrical double layer (EDL) at the gate/electrolyte/semiconductor interfaces [35]. Since EDL acts as an efficient gate dielectric, a drastic reduction in operating voltages is essentially guaranteed as the typical EDL capacitance ranges from a few to hundreds of µF/cm^2^ compared to nF/cm^2^ for standard dielectrics [36]. Despite these advantages of bio-FETs, several unresolved problems still hinder their technological transfer, such as their electrical stability. As a consequence, it is extremely important to develop a reliable and efficient device that can be transferred to the electronic market for mass production.

In this work, we fabricated and studied thin-film electrolytic transistors, using Al:ZnO as a biosensor. Due to the high chemical activity of ZnO, deionized water was used as the gate dielectric since electrolytes, such as PBS (phosphate buffered saline), can chemically react with the transistor channel material, thereby disrupting the correct operation of the device. The novelty of this study lies in the application of electrochemical methods to establish the operating range of the device. Cyclic voltammetry makes it possible to determine the size of the electrochemical window for transistor materials and, as a result, the values of the maximum allowable operating voltages on transistor elements. Electrochemical impedance spectroscopy was used as an effective tool to determine the electrical capacitance of the gate–channel system. The techniques demonstrated in this work can be applied to optimize the operating parameters of various semiconductor materials in order to create a universal detection platform for biosensing applications, such as multi-element FET sensor arrays based on nanostructured films of various compositions, which use neural network signal processing.

## 2. Materials and Methods

Polyamide substrates (15 × 15 mm in size and 0.15 mm thick) were fabricated by laser cutting. They were then washed in isopropanol (CAS number: 67-63-0) in an ultrasound bath for 10 min. The contact masks for sputtering were made of 0.2 mm-thick stainless steel (AISI 304) sheet using a laser demetallization technique on a TruMark 3000 machine. TFT electrodes were fabricated by the magnetron sputtering of a four-inch chromium target (CAS number: 7440-47-3) in an argon flow of 30 sccm with 1000 W power. Next, a Ta_2_O_5_ passivation layer was deposited by the reactive sputtering of a four-inch tantalum metal target (CAS number: 7440-25-7) in an Ar/O_2_ medium at a gas ratio of 20:100 sccm and at a discharge power of 500 W. An Al:ZnO semiconductor layer was also deposited by the magnetron sputtering of a five-inch Al:ZnO target (1:99 ratio) (CAS number: 7440-25-7) at a sputtering power of 200 W. Throughout every step of sample preparation, the distance between the target and the magnetron was 25 cm.

X-ray photoelectron spectroscopy (XPS) measurements were performed using an ESCALAB Xi (Thermo Fisher Scientific Inc., Waltham, MA, USA) X-ray photoelectron spectrometer to verify the chemical composition of the deposited Al:ZnO films. The morphological aspects of the films were assessed using a TESCAN MAIA3 TRIGLAV (Tescan, Brno, The Czech Republic) scanning electron microscope (SEM) and a PARK NX10 (Park Systems, Suwon, Korea) atomic force microscope (AFM) in non-contact mode.

Measurements of the size of the electrochemical window and the capacitance of the gate–channel system were carried out with a Zahner Zennium (Zahner-Elektrik GmbH & Co., Kronach – Gundelsdorf, Germany) electrochemical station, using the method of cyclic voltammetry with a voltage range of ±2.0 V and a scanning speed of 50 mV/s. The electrical capacitance of the gate–semiconductor system was determined using impedance spectroscopy in a range of frequencies from 1 Hz to 1 MHz at a signal amplitude of 0.5 V. The current–voltage characteristics of the transistor were investigated using a setup consisting of two Keithley 2400 (Keithley Instruments, LLC, Solon, Ohio USA) source-meter units. Single-stranded OPE-01 DNA primers obtained from the BIONEER (Bioneer Inc., Daejeon, Korea) company (10 mer long (CCC AAG GTC C) and with a molar mass of 2972.9 g/mol) were used as an analyte at four different concentrations ranging from 1 to 1000 pM/µL. All measurements were carried out at an ambient temperature of 21 °C and a humidity of 35%.

## 3. Results and Discussion

Figure 1b shows a photograph of an actual polyimide substrate with an array of electrolyte-gated transistors. A schematic of a single device unit, including all the components, is shown in Figure 1c. SEM was used to image the as-prepared devices at high resolution (see Figure 1d). The fabricated device had the following geometric properties: the thickness of the metal electrodes (drain, source, gate) was 25 nm, the Ta_2_O_5_ passivation layer was 40 nm thick, the Al:ZnO layer was 50 nm thick, the length of the transistor channel was 170 μm and its width was 2100 μm, the gate electrode area was 1.83 mm^2^, the area of the Al:ZnO layer was 1.5 mm^2^, and the distance between the gate and the semiconductor layer was 150 µm.

An XPS analysis was performed on the Al:ZnO thin films in order to verify their elemental composition and to study the chemical states of the component ions in the film surface. The compositional analysis confirmed the presence of slightly sub-stoichiometric ZnO (oxygen vacancies), as well as around 1% Al content and no other impurities. High-resolution spectra were obtained for Zn 2p, O 1s, and Al 2p regions in order to understand the chemical nature of the elements. The spectra were calibrated relative to the adventitious C 1s peak at 284.8 eV. The Zn 2p scan (Figure 2a) shows the presence of two Zn 2p_3/2_ and 2p_1/2_ peaks with a spin–orbit splitting of 23.1 eV and an area ratio of 2:1. The Zn 2p_3/2_ peak is located at 1021.5eV, which is concordant with the Zn 2+ oxidation state in ZnO [37]. The O 1s scan is depicted in Figure 2b, and a peak containing two components can be distinguished. The peak at 529.9 eV can be attributed to the O 1s state in wurtzite ZnO. The component centered around 531.5 eV is typically attributed to organic contaminations (C-O chemical state) on the surface; however, a contribution at the same binding energy might arise from an oxygen-deficient region within the ZnO matrix [37]. Furthermore, the Al 2p scan (Figure 2c) was measured in order to determine the chemical nature of the Al dopant ions. A peak centered at approximately 73.5 eV is commonly attributed to AlO_x_ (Al-O-Zn bonds) in Al:ZnO films [38]. Such bonding is typical for 1% Al in ZnO, since the Al content is not high enough to create fully oxidized Al_2_O_3_-based defect complexes.

A surface study using non-contact AFM (Figure 3) revealed a rather rough, grainy surface with grain sizes ranging from 30 to 100 nm, although the film itself was solid and uniform. A rough surface has an advantage compared to a smooth surface in that it has a higher surface area. This results in a higher electric capacitance, which allows for more control over FET operation. However, such surfaces can exhibit high hydrophobic effects that may significantly affect the readings in the presence of water-based electrolytes.

One of the most significant characteristics of FET with an electrolytic gate is the electrochemical stability and capacitance of the gate–channel system, which is important for the subsequent calculation of the electrical properties of the transistor channel’s semiconductor material. The values of electrochemical stability are known as the electrochemical window, indicating the range of potentials in which the electrolyte is neither oxidized nor reduced. The results from cyclic voltammetry in Figure 4 clearly indicate the approximate size of the electrochemical window as ranging from −0.6 V to +1.2 V (Figure 4a), which determines the maximum value of the applied voltage at the transistor gate. Exceeding these values will lead to the electrochemical doping of the transistor channel, which will greatly affect the electrical output characteristics of the transistor, thus complicating the identification of the analyte signal.

In order to determine the gate–channel electrical capacitance, impedance spectroscopy was applied in a frequency range from 1 Hz to 1 MHz with a scan duration of 2 min. The results are shown in Figure 4b. A pronounced diffusion region (Warburg diffusion) was observed in the frequency range 1–80 Hz. The thickness of the diffuse region of the electrical double layer depends on the ion concentration in the solution: the lower the ion concentration, the thicker the diffuse layer [39]. According to our calculations using these data, the capacitance of the system was 40 nF/cm^2^, which was much lower than expected. These low values for the electrical capacitance can be explained based on the fact that pure water itself is a weak electrolyte, and the main roles in the formation of EDL are played by the products of the self-ionization of water (hydronium ions H_3_O^+^, protons H^+^ and hydroxide ions OH^−^), rather than the dissociation products of impurities (layers of metals, acids) [40].

Voltage ranges for the electrical characterization were established based on the results of the electrochemical measurements (Figure 4a). The output characteristic (I_d_–V_d_) was measured for a drain-source voltage range of 0–1 V with a step of 0.01 V, time intervals of 50 ms between each data point, and a gate voltage of 0 to +0.7 V with a step of 0.1 V. The total acquisition time for each curve was 5 s (Figure 5a). The obtained output characteristics are typical for n-type devices demonstrating well observable linear and saturation regions of the transistor channel. The transfer characteristic (I_d_–V_g_) was determined for a range of gate voltage from 0 to +1. V with a step of 0.01 V, and for drain-source voltages of 0.1 to 0.5 V with a step of 0.01 V. This particular FET device had an ON/OFF ratio > 10^3^ at a gate potential of 1 V. A drain voltage above 0.5 V leads to strong channel saturation and Vt drift to the negative side (Figure 5a,d), both of which are common phenomena in such systems. In the case of magnetron sputtered Al:ZnO, the samples have much lower porosity and higher density, which has a positive effect on the long-term stability of the samples. The maximum time between the manufacturing of transistors and the reporting of their characteristics was 5 days. During this time, no changes in the working properties of the samples were observed.

A low drain-source voltage value of 10 mV was used to minimize the threshold voltage drift to the negative side due to channel saturation. The output curves (Figure 5a) for the WGFET and the results of the ELR method (Figure 5c) gave a threshold voltage value of 654 mV. The value of the field mobility was calculated based on the dimensions of the transistor, the capacitance, and the drain current in saturation mode using the following expression:$$I_{d}={µC}_{i}\frac{W}{2L}{\left(V_{\mathrm{gs}}-V_{\mathrm{th}}\right)}^{2},$$
where $I_{d}$ is the drain current, µ is the field-effect mobility, $C_{i}$ is the gate–channel capacitance per unit area, W is the transistor channel width, L is the channel length, $V_{\mathrm{gs}}$ is the gate voltage, and $V_{\mathrm{th}}$ is the threshold voltage.

Based on the above values for the capacitance and saturation current, the charge carrier mobility in the transistor channel material was calculated as 6.85 cm^2^/(V s), a value that is consistent with the results of similar studies [41,42].

The next set of measurements were carried out to study of the effects of the DNA aptamers on the conductivity of the transistor channel (Figure 6). The objects of study were the above-mentioned aptamers, diluted with deionized water at four different concentrations. All measurements were carried out at a gate potential of +1.0 V and a drain-source voltage of 0.4 V for the transistor, in accordance with the above-mentioned current–voltage characteristics for pure water.

Increasing the concentration of the DNA aptamers resulted in a decrease in the conductivity of the channel compared to the reference analyte (DI water). The relative change in the conductivity of the FET channel was 14.8% at maximum concentration compared to the reference. The isoelectric point (IEP) for ZnO lies within a pH range of 8.0–10.0 [43], while the IEP for the DNA/RNA molecule lies within a pH range of 4.35–6.7 [44], resulting in a strong negative net charge due to its phosphate backbone. This results in a strong electrostatic attraction between the positively charged ZnO surface and the negatively charged aptamer, thereby inhibiting the conductivity of the n-type channel of the transistor. Another important aspect is that DNA biopolymers have the properties of dielectrics, with permittivity in the range of 3–100 units [45,46] depending on the measurement conditions. The O and N atoms have negative charges, while the H atom has a positive charge. In addition, there are two types of chemical bonds in double-stranded DNA: the OH bond (O from the thymine molecule and H from the adenine molecule) and the NH-N bond (NH from the thymine molecule and N from adenine). The distance between the atoms in each molecule has previously been calculated as 1.36 Å nm [47]. Thus, two strands of double-stranded DNA are held together by electrostatic forces due to the net average charge between the H and N atoms, and between the C and O atoms, calculated as 0.2e and 0.4e, respectively. In a DNA suspension, single-stranded/double-stranded DNA molecules are distributed randomly, and an electric field can be applied to polarize these strands. The intrinsic properties of the DNA strands determine the strength of polarization, which is a measure of its ability to hold electrical charge. In addition, DNA may be in different phase states [48], such as an isotropic liquid in an aqueous solution, a liquid-crystalline phase (nematic, cholesteric), or a crystal. It is known that negatively charged DNA helices in aqueous solutions can condense [49] due to electrostatic interaction in the presence of positively charged counter-ions in the solution, which accumulate in the grooves of the double helices [50]. The combination of these factors creates a rather complex and integral picture of the processes occurring on the surface of the biosensor.

Table 1 provides a summary of the properties of the various types of EGFETs with different electrolytes, as reported previously. According to the results of the comparative analysis, it can be seen that the electrical capacitance of the system for different types of transistors lies in the range from 0.9 to 6 μF cm^−2^ and is maximum in the case of polymer electrolytes. In turn, the greatest mobility of charge carriers and the best ON/OFF ratio are achieved by using inorganic semiconductor materials, such as the transistor channel.

## 4. Conclusions

In this work, we demonstrated a design for, and the optimal parameters of, an FET biosensor based on Al:ZnO. Good electric properties, low-temperature processing, and low operating voltage make this type of device promising for use in biosensor applications on temperature-sensitive substrates. In addition, the polymer substrate makes the application of this type of FET device possible in the field of flexible electronics. Critical electrical measurements were made to determine the performance of this sensor. Electrochemical impedance spectroscopy and cyclic voltammetry were used to accurately establish parameters, such as the capacitance of the gate–channel system and the size of the electrochemical window, which were necessary to determine the operating voltage range of the transistor. The geometric features of the transistor presented in this research could be adapted to study the electrical properties of both organic and inorganic semiconductor materials in an EGFET configuration. Al:ZnO itself is a promising material for biosensors using thin-film transistors due to the high isoelectric point and good electric properties. However, the high chemical reactivity of this material limits its application in a relatively inert environment. Another way to increase the sensitivity of such devices is to use nanostructured coatings, which can significantly increase the working surface area of the sensor. This work sets the ground for next-generation biosensor devices with advanced selectivity and sensitivity, including a platform of multi-element FET sensor arrays based on nanostructured films of various compositions with functionalized surfaces using neural network signal processing. This may enable the study of analytes of complex composition with a high-precision response.

## Figures and Tables

**Figure 1 sensors-22-03408-f001:**
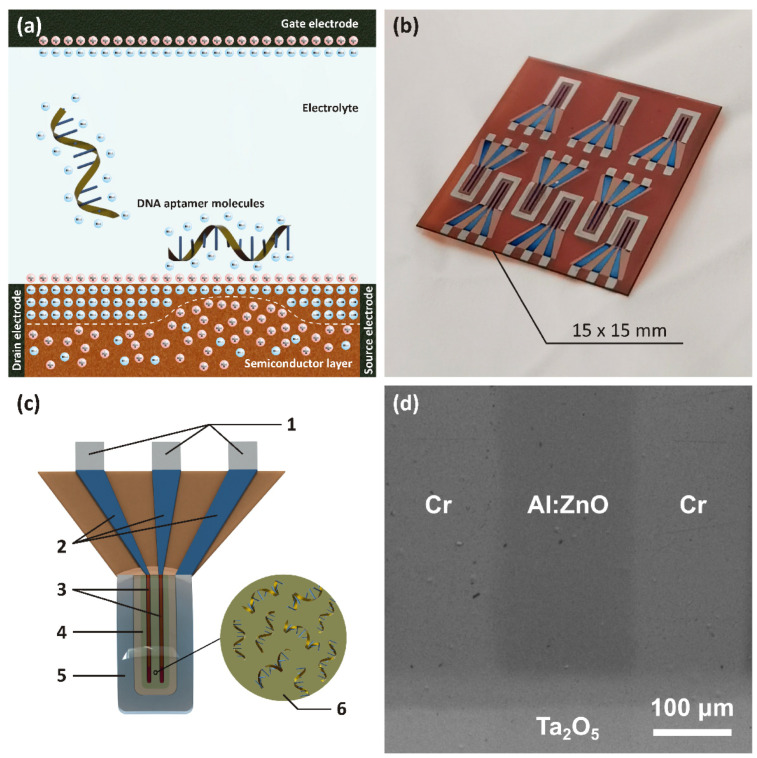
(**a**) Schematic representation of the principle of operation of bio-FET. (**b**) A photograph of the actual polyimide substrate hosting an array of 9 WGFETs. (**c**) A schematic of a single device unit: 1—gate, source and drain contact pads; 2—gate, source and drain electrodes passivated with Ta_2_O_5_; 3—Al:ZnO layer on top of the drain-source channel; 4—Al:ZnO layer; 5—the gate electrode, 6—DNA aptamers attached to the Al:ZnO surface. (**d**) SEM image of the transistor channel dimension measurements.

**Figure 2 sensors-22-03408-f002:**
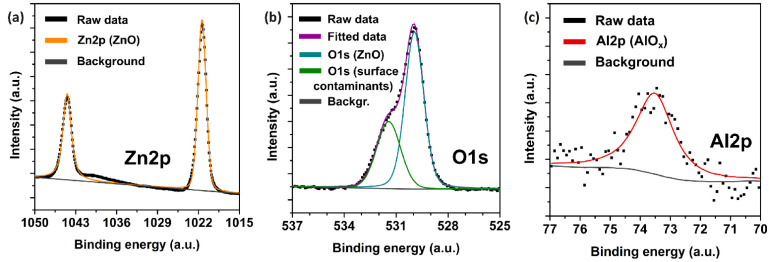
High-resolution XPS spectra and peak fits of the Al:ZnO thin film: (**a**) Zn 2p scan, (**b**) O 1s scan, and (**c**) Al 2p scan.

**Figure 3 sensors-22-03408-f003:**
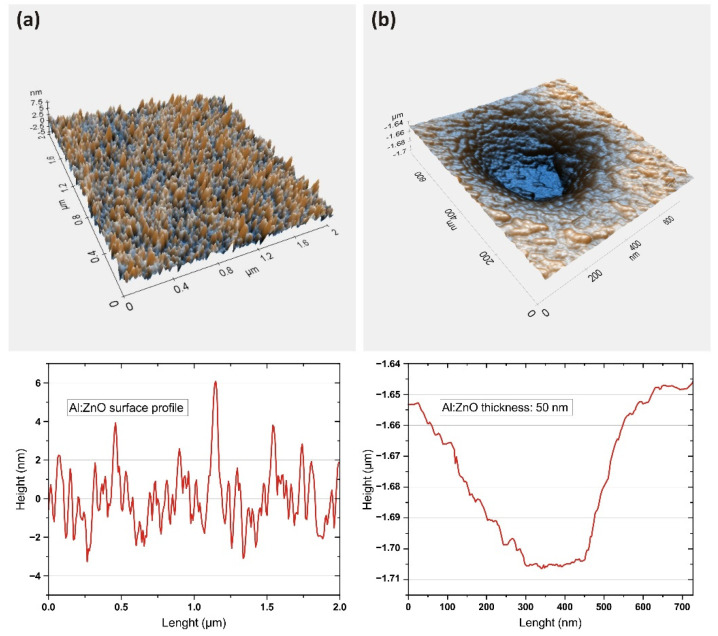
(**a**) Anisotropic AFM image of the room-temperature magnetron-sputtered Al:ZnO layer surface profile; (**b**) thickness measurements.

**Figure 4 sensors-22-03408-f004:**
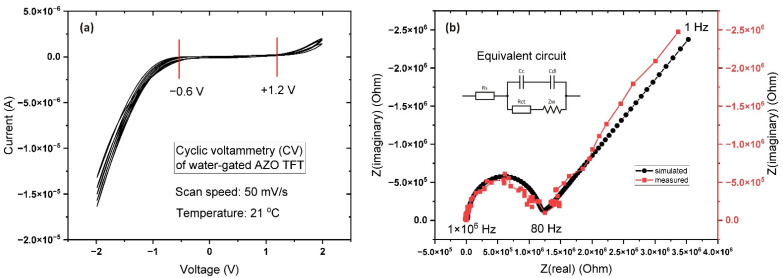
(**a**) Cyclic I−V curve for DI water in contact with a metal/semiconductor (red vertical lines indicate the electrochemical window for the particular device (chromium/DI water/Al:ZnO) that is equal to 1.8 V); (**b**) Nyquist plot of the impedance spectra for Al:ZnO EGFET and the equivalent electrical circuit. An AC signal is applied between the gate electrode and the shorted together drain-source electrodes.

**Figure 5 sensors-22-03408-f005:**
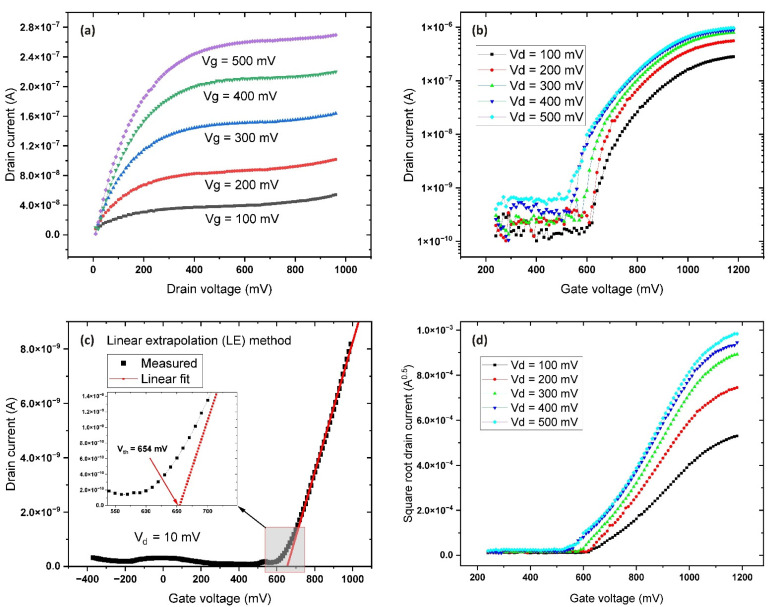
(**a**) Output characteristics (I_d_–V_d_) of the WGFET at gate voltages ranging from 100 mV to 500 mV and drain voltages from 0 V to 1 V; (**b**) logarithmic representation of transfer characteristics (I_d_–V_g_) of the thin-film transistor; (**c**) results from the extrapolation of the linear region (ELR) method for determination of V_th_; (**d**) square root plot of the drain current vs. gate voltage.

**Figure 6 sensors-22-03408-f006:**
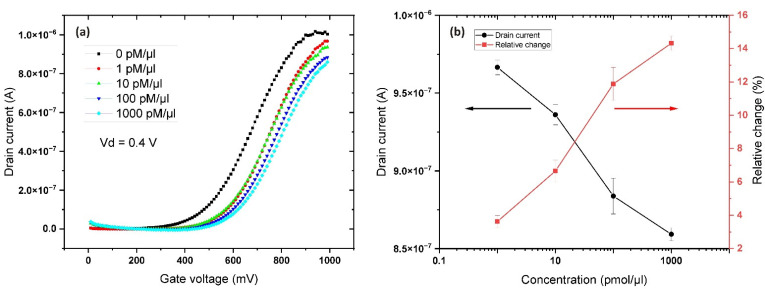
(**a**) Transfer (I_d_−V_g_) curves at 0.4 V drain voltage for aptamer containing analyte at concentrations of 1–1000 pM/μL, and (**b**) the relative sensor response level.

**Table 1 sensors-22-03408-t001:** A summary of the properties of the various types of EGFETs.

Dielectric Material (Electrolyte)	Semiconductor Material	Configuration	Electrode Material (D&S/G)	V_th_	I_ON_/I_OFF_	μ (cm^2^ V^−1^ s^−1^)	C (μF cm^−2^)	Ref.
(DEME) TFSI	MoS_2_	In-plane	Ti/Au	0.5	10^7^	60	1.55	[51]
LiClO4, PVA, PC	In_2_O_3_	In-plane	ITO/ITO	N/A	10^6^	98.3	5.97	[52]
Purified water	P3HT	TGBC	Au/Au	−0.16	150	5.9 × 10^−3^	3	[53]
PVA, PEMA, DMSO	In_2_O_3_	TGBC	NR/PEDOT:PSS	−0.138	1.3 × 10^6^	N/A	5.4	[54]
DI water	TIPS-pentacene	TGBC	Au/Pt wire	−0.140	100	1.3 × 10^−2^	3.8	[55]
138 mM NaCl	TIPS-pentacene	TGBC	Au/Pt wire	0.05	100	1.7 × 10^−3^	1.75	[55]
0.2 M NaCl	PBTTT (non-annealed)	TGBC	Au/W tip	0	100	8 × 10^−2^	0.9	[56]
DI water	Al:ZnO	In-Plane	Cr/Cr	0.654	3 × 10^3^	6.85	0.04	This work

## Data Availability

Not applicable.

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
