# Peer review of "Effect of DNA Aptamer Concentration on the Conductivity of a Water-Gated Al:ZnO Thin-Film Transistor-Based Biosensor"

_sensors, 2022, doi:10.3390/s22093408_

Round 1
Reviewer 1 Report
The authors present the effects of DNA aptamer concentration on the conductivity of a water-gated Al:ZnO TFT-based biosensors. The authors deposited the Al:ZnO TFT channel via sputtering and studied the aptamer dependent change in the electrical properties of the device. The research idea is well conceived; however, there are many shortcomings that need serious attention for publication. Hence, I would like to decide based on the major revision of the following comments.
1- Explain the novelty and the exigence of the presented research idea and study in the abstract.
2- The first three paragraphs of the introduction section provides too general information on biosensors. I recommend merging the most important information into a single paragraph and provide a relevant technical information on the subject with a recent literature review.
3- Provide the different methods of ZnO fabrication and the reason why the authors preferred sputtering over these methods. Citing the following papers of different ZnO fabrication methods would be useful for the readers.
a. Yang, P.; Yan, H.; Mao, S.; Russo, R.; Johnson, J.; Saykally, R.; Morris, N.; Pham, J.; He, R.; Choi, H.J. Controlled growth of ZnO nanowires and their optical properties. Adv. Func. Mater. 2002, 12, 323–331.
b. Park, W. I.; Yi, G.C.; Kim, M.; Pennycook, S.J. ZnO Nanoneedles Grown Vertically on Si Substrates by Non‐Catalytic Vapor‐Phase Epitaxy. Adv. Mater. 2002, 14, 1841–1843.
c. Shaikh, S.F.; Al-Enizi, A.M.; Agyeman, D.A.; Ghani, F.; Nah, I.W.; Shahid, A. Intrinsic control in defects density for improved ZnO nanorod-based UV sensor performance. Nanomaterials, 2020, 10, 142.
d. Sun, Y.; Fuge, G.M.; Ashfold, M.N. Growth of aligned ZnO nanorod arrays by catalyst-free pulsed laser deposition methods. Chem. Phys. Lett. 2004, 396, 21–26.
e. Yao, B.D.; Chan, Y.F.; Wang, N. Formation of ZnO nanostructures by a simple way of thermal evaporation. Appl. Phys. Lett. 2002, 81, 757–759.
4- Provide the CAS numbers of all the chemicals used in the experiments.
5- The Al:ZnO is wrongly written as Al:ZO many times in the manuscript. Double check the manuscript for typos and errors.
6- Label the top silver surface in Figure 1(b). Also, term the EDS data as Figure 1(d) for clarity.
7- In the results section, the authors have not qualify the fact that why did they use the Al-doped ZnO instead of naturally n-type intrinsic ZnO? Draw a relation between Al-doing and aptamer DNA concentration and why is it preferred over intrinsic ZnO? Secondly, why did the authors prefer Al-doping over other metallic doping into ZnO?
8- For readers' understanding, compare the efficiency of the presented device with the efficiencies of the devices propounded in recently published literature in a tabular form.
9. The znO electrical properties are directly dependent on the surface and deep level defects in the ZnO structure. The authors did not report the defect states in the sputtered ZnO and the effect of doping on those defects? The study of the electrical properties of ZnO is incomplete without reporting the defect states.
10. With the help of a schematic Figure, explain the sensing mechanism of the presented DNA aptamer biosensor?
11- In the conclusion section, explain the limitations to this study and the future directions required to improve the technology?
Reviewer 2 Report
The paper presented the effect of DNA aptamer concentration on the water-gated AZO TFT biosensor’s performance. Lack of an essential description of the state-of-the art research is the main problem of the work. From this point of view, the authors have given no clue on what is new and what was achieved in this work.
Another question is the confusing motivation. As mentioned in the introduction part, ‘several unresolved problems still hinder their technological transfer, such as their electrical stability’. However, there is no any result on this aspect.
Considering these deficiencies, the major scientific points are missing, which makes it hard to meet the criteria of this journal.
Round 2
Reviewer 1 Report
The authors have provided the response to the review comments and the manuscript has been improved a lot. However, I still believe that there are certain shortcomings in the manuscript that need serious attention for publication. I want the authors to consider the following comments for publication.
1- The added information in the abstract is simply the introduction of the FET-based bio sensors. I was asking the authors to provide the specific novelty of their work in the manuscript that makes it a significant contribution to science? I believe the same was suggested by the reviewer 2.
2- The introduction section is still very long and descriptive?
3- ZnO is naturally an n-type material because of the presence of donor oxygen vacancies. If the Al ions act as oxygen vacancies suppressors than how could the carrier concentration be increased? Isn't is remains the same as an Al ion is suppressing an oxygen vacancy and supplanting an electron. How would the carrier concentration be increased? Explain in the manuscript.
4- The authors could study the defect levels via room temperature photoluminescence and XPS studies if DLTS is a challenge? I believe the same is suggested by the reviewer 2.
Reviewer 2 Report
The authors carefully considered my previous comments, and made corresponding revisions. I think it is acceptable in present form.
Round 3
Reviewer 1 Report
The authors have sufficiently addressed all the concerns. The manuscript now looks good and viable for Sensors. I would like to accept the manuscript in the present form.
